# GradAug: A New Regularization Method for Deep Neural Networks

**Taojiannan Yang, Sijie Zhu, Chen Chen**
University of North Carolina at Charlotte
{tyang30,szhu3,chen.chen}@uncc.edu

## Abstract

We propose a new regularization method to alleviate over-fitting in deep neural networks. The key idea is utilizing randomly transformed training samples to regularize a set of sub-networks, which are originated by sampling the width of the original network, in the training process. As such, the proposed method introduces self-guided disturbances to the raw gradients of the network and therefore is termed as Gradient Augmentation (GradAug). We demonstrate that GradAug can help the network learn well-generalized and more diverse representations. Moreover, it is easy to implement and can be applied to various structures and applications. GradAug improves ResNet-50 to **78.79%** on ImageNet classification, which is a new state-of-the-art accuracy. By combining with CutMix, it further boosts the performance to **79.67%**, which outperforms an ensemble of advanced training tricks. The generalization ability is evaluated on COCO object detection and instance segmentation where GradAug significantly surpasses other state-of-the-art methods. GradAug is also robust to image distortions and FGSM adversarial attacks and is highly effective in low data regimes. Code is available at `https://github.com/taoyang1122/GradAug`

## 1 Introduction

Deep neural networks have achieved great success in computer vision tasks such as image classification [1,2], image reconstruction [3,4], object detection [5,6] and semantic segmentation [7,8]. But deep neural networks are often over-parameterized and easily suffering from over-fitting. Regularization [9,10] and data augmentation [1,11] are widely used techniques to alleviate the over-fitting problem. Many data-level regularization methods [10,12,13] have achieved promising performance in image classification. These methods are similar to data augmentation where they put constraints on the input images. Although effective in image classification, these methods are hard to apply to downstream tasks such as object detection and segmentation due to their special operations. For example, the state-of-the-art CutMix [13] can not be directly applied to object detection because first, mixing samples will destroy the semantics in images; second, it is hard to interpolate the labels in these tasks. Another category of regularization methods imposes constraints on the network structures. [14] proposes that adding noises to the network gradients can improve generalization. Other methods [9,15,16] randomly drop some connections in the network, which implicitly introduce random noises in the training process. These methods are usually more generic but not as effective as data-level regularization.

In this paper, we introduce Gradient Augmentation (GradAug), which generates meaningful disturbances to the gradients by the network itself rather than just adding random noises. The idea is that when a random transformation (e.g., random rotation, random scale, random crop, etc.) is applied to an image, a well-generalized network should still recognize the transformed image as the same object. Different from the regular data augmentation technique which only regularizes the full-network, we regularize the representations learned by a set of sub-networks, which are randomly sampled

from the full network in terms of the network width (i.e., number of channels in each layer). Since the representation of the full network is composed of sub-networks' representations due to weights sharing during the training, we expect sub-networks to learn different representations from different transformations, which will lead to a well-generalized and diversified full network representation.

We conduct a comprehensive set of experiments to evaluate the proposed regularization method. *Using a simple random scale transformation*, GradAug can improve the ImageNet Top-1 accuracy of ResNet-50 from 76.32% to **78.79%**, which is a new state-of-the-art accuracy. By leveraging a more powerful data augmentation technique – CutMix [13], we can further push the accuracy to **79.67%**. The representation's generalization ability is evaluated on COCO object detection and instance segmentation tasks (Section 4.4). Our ImageNet pretrained model alone can improve the baseline MaskRCNN-R50 by **+1.2** box AP and **+1.2** mask AP. When applying GradAug to the detection framework, it can outperform the baseline by **+1.7** box AP and **+2.1** mask AP. Moreover, we demonstrate that GradAug is robust to image corruptions and adversarial attacks (Section 4.5) and is highly effective in low data settings (Section 4.6).

## 2    Related Work

**Data augmentation.** Data augmentation [1, 11, 17] increases the amount and diversity of training data by linear or non-linear transformations over the original data. In computer vision, it usually includes rotation, flipping, etc. Recently, a series of regularization methods use specially-designed operations on the input images to alleviate over-fitting in deep neural networks. These methods are similar to data augmentation. Cutout [10] randomly masks out a squared region on the image to force the network to look at other image context. Dropblock [18] shares a similar idea with Cutout but it drops a region in the feature maps. Although they have achieved improvements over the regular data augmentation, such region dropout operations may lose information about the original images. Mixup [12] mixes two samples by linearly interpolating both the images and labels. CutMix [13] combines Cutout and Mixup to replace a squared region with a patch from another image. Other mixed sample variants [19, 20] all share similar ideas. While effective in image classification, the mixed sample augmentation is not natural to be applied to tasks such as detection and segmentation due to semantic and label ambiguities. In contrast, the proposed GradAug is a task-agnostic approach which leverages the most common image transformations to regularize sub-networks. This allows the method to be directly applied to different vision tasks and easily amenable for other applications.

**Structure regularization.** Another category of regularization methods imposes constraints on the network weights and structure to reduce over-fitting. [14] points out that adding random noises to the gradients during training can help the network generalize better. Dropout [9] randomly drops some connections during training to prevent units from co-adapting. The random dropping operation also implicitly introduces random noises into the training process. Many following works share the idea of Dropout by randomly dropping network layers or branches. Shake-Shake [21] assigns random weights to residual branches to disturb the forward and backward passes. But it is limited to three-branch architectures. ShakeDrop [22] extends Shake-Shake to two-branch architectures (e.g., ResNet [2] and PyramidNet [23]). However, its application is still limited. [15] randomly drops a subset of layers during training. The final network can be viewed as an ensemble of many shallow networks. Although these methods have shown improvements on image classification, they are usually not as effective as data-level regularization strategies. Moreover, their generalization and effectiveness are not validated on other tasks.

GradAug leverages the advantages of both categories of methods. It uses different augmentations to regularize a set of sub-networks generated from the full network in the joint training process. This introduces self-guided disturbances to the gradients of the full network rather than adding random noises. The method is more effective and generic than previous techniques.

## 3    GradAug

### 3.1    Algorithm

When applying some random transformations to an image, human can still recognize it as the same object. We expect deep neural networks to have the same generalization ability. GradAug aims to regularize sub-networks with differently transformed training samples. There are various of

methods to generate sub-networks during training. Previous works [9, 15, 16] usually stochastically drop some neurons, layers or paths. In GradAug, we expect the final full-network to take advantage of the learned representations of the sub-networks. Therefore, we sample sub-networks in a more structured manner, that is by the network width. We define $\theta$ as the model parameter. Without loss of generality, we use convolutional layers for illustration, then $\theta \in \mathbb{R}^{c_1 \times c_2 \times k \times k}$, where $c_1$ and $c_2$ are number of input and output channels, $k$ is the convolution kernel size. We define the width of a sub-network as $w \in [\alpha, 1.0]$, where $\alpha$ is the width lower bound. The weights of the sub-network is $\theta_w$. Different from random sampling, we always sample the first $w \times 100\%$ channels of the full-network and the sub-network weights are $\theta_w \in \mathbb{R}^{wc_1 \times wc_2 \times k \times k}$.

In this way, a larger sub-network always share the representations of a smaller sub-network in a weights-sharing training fashion, so it can leverage the representations learned in smaller sub-networks. Iteratively, sub-networks can construct a full-network with diversified representations. Figure 1 shows the class activation maps (CAM) [24] of the sub-network and full-network. The full-network pays attention to several regions of the object because it can leverage the representation of the sub-network. For example, when the sub-network ($w = 0.9$) focuses on one dog in the image, the full-network shares this attention and uses the other network part to capture the information of another dog. Therefore, the full-network learns richer semantic information in the image, while the baseline model only models a single region and does not fully comprehend the salient information of the image. To make the method

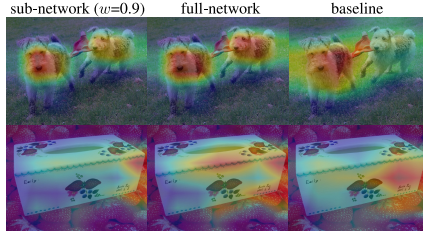

Figure 1: Class activation maps (CAM) of the network trained by GradAug and the baseline. The full-network shares the attention of the sub-network and focuses on multiple semantic regions.

simple and generic, we choose among the most commonly used transformations such as random scales, random rotations, random crops, etc. In the experiments, we show that a simple random scale transformation can already achieve state-of-the-art performance on image classification, and it can be directly applied to other applications. Moreover, we can use more powerful augmentations such as CutMix for further enhanced performance.

**Training procedure.** The training procedure of GradAug is very similar to the regular network training. In each training iteration, we train the full-network with the original images, which is the same as the regular training process. Then we additionally sample $n$ sub-networks and train them with randomly transformed images. Finally, we accumulate the losses of full-network and sub-networks to update the model weights. This naive training approach achieves good training accuracy but the testing accuracy is very low. This is caused by the batch normalization (BN) [25] layers. The BN layer will collect a moving average of training batches' means and variances during training. The collected mean and variance will be used during inference. However, the batch mean and variance in the sub-networks can be very different from those in the full-network because the training samples are randomly transformed. This will cause the final BN mean and variance to be inappropriate for the full-network during inference. But in the training phase, BN uses the mean and variance of the current batch, so the training behaves normally. To obtain the correct BN statistics for the full-network, *we do not update BN mean and variance when training the sub-networks*. Only the full-network is allowed to collect these statistics. However, the weights in BN layer are still updated by sub-networks because they can be shared with full-network. To further improve the performance, we also leverage two training tricks in [26]. First, we use the output of the full-network as soft labels to train the sub-networks. Second, we always sample the smallest sub-network (i.e., $w = \alpha$) during training if $n > 1$. The effect of these two training tricks is provided in the **supplementary material**. The Pytorch-style pseudo-code of GradAug is presented in Algorithm 1.

## 3.2 Analysis of gradient property

We provide an in-depth analysis of GradAug from the perspective of gradient flow. For simplicity, we consider a fully connected network with 1-D training samples. We define the network as $N$. The parameter of one layer in the full-network is $\theta \in \mathbb{R}^{c_1 \times c_2}$. The parameter of sub-networks is $\theta_w$ as explained in Section 3.1. $x \in \mathbb{R}^d$ is the training sample and $y$ is its label. The output of the network is denoted as $N(\theta, x)$, and the training loss is $l(N(\theta, x), y)$ where $l$ is the loss function, which is often the cross entropy in image classification. The loss and gradients in a standard training process

---

**Algorithm 1** Gradient Augmentation (GradAug)

---
**Input:** Network $Net$. Training image $img$. Random transformation $T$. Number of sub-networks $n$. Sub-network width lower bound $\alpha$.
▷ Train full-network.
Forward pass, $output_f = Net(img)$
Compute loss, $loss_f = criterion(output, target)$
▷ Regularize sub-networks.
**for** $i$ in $range(n)$ **do**
    Sample a sub-network, $subnet_i = Sample(Net, \alpha)$
    Fix BN layer's mean and variance, $subnet_i.track\_running\_stats = False$
    Forward pass with transformed images, $output_i = subnet_i(T^i(img))$
    Compute loss with soft labels, $loss_i = criterion(output_i, output_f)$
**end for**
Compute total loss, $L = loss_f + \sum_{i=1}^{n} loss_i$
Compute gradients and do backward pass

---

are computed as

$$L_{std} = l(N(\theta, x), y), \quad g_{std} = \frac{\partial L_{std}}{\partial \theta}, \tag{1}$$

where $g_{std} \in \mathbb{R}^{c_1 \times c_2}$. Structure regularization methods [9, 15, 16] randomly drop some connections in the network, and their loss and gradients can be computed as

$$L_{sr} = l(N(\theta_{rand}, x), y), \quad g_{sr} = \frac{\partial L_{sr}}{\partial \theta_{rand}}. \tag{2}$$

We can view $g_{sr}$ has the same shape as $g_{std}$ where the gradients of disabled connections are 0. Therefore, we can rewrite $g_{sr}$ as

$$g_{sr} = g_{std} + g_{noise}, \tag{3}$$

where $g_{noise} \in \mathbb{R}^{c_1 \times c_2}$ is a random matrix which introduces some random disturbances to the gradients. In contrast, GradAug applies more meaningful disturbances to the gradients. Let $T$ be the random transformation operation (e.g., random scale, random rotation, etc.) and $T^i$ be the transformation to sub-network $i$ ($i = [1, ..., n]$). The loss and gradients are computed as:

$$L_{GA} = l(N(\theta, x), y) + \sum_{i=1}^{n} l(N(\theta_{w_i}, T^i(x)), N(\theta, x))$$

$$g_{GA} = \frac{\partial l(N(\theta, x), y)}{\partial \theta} + \sum_{i=1}^{n} \frac{\partial l(N(\theta_{w_i}, T^i(x)), N(\theta, x))}{\partial \theta_{w_i}} = g_{std} + g'. \tag{4}$$

$g_{GA}$ has a similar form with $g_{sr}$. The first term is the same as the gradients in standard training. *But the second term $g'$ is derived by the sub-networks with transformed training samples. Since sub-networks are part of the full-network, we call this term "self-guided".* It reinforces good descent directions, leading to improved performance and faster convergence. $g'$ can be viewed as an augmentation to the raw gradients $g_{std}$. It allows different parts of the network to learn diverse representations.

The gradients of data-level regularization methods are similar to $g_{std}$, with the difference only in the training sample. The gradients are

$$g_{dr} = \frac{\partial l(N(\theta, f(x)), y)}{\partial \theta}, \tag{5}$$

where $f$ is the augmentation method such as CutMix. GradAug can also leverage these augmentations by applying them to the original samples and then following random transformations. The gradients become

$$g_{GA} = \frac{\partial l(N(\theta, f(x)), y)}{\partial \theta} + \sum_{i=1}^{n} \frac{\partial l(N(\theta_{w_i}, T^i(f(x))), N(\theta, f(x)))}{\partial \theta_{w_i}} = g_{dr} + g'. \tag{6}$$

$g'$ is still an augmentation to $g_{dr}$. Data augmentation can also be combined with other structure regularization methods. However, similar to the derivations in Eq. 2 and Eq. 3, such combination strategy introduces random noises to $g_{dr}$, which is not as effective as GradAug as shown in Table 3.

Table 1: ImageNet classification accuracy of different techniques on ResNet-50 backbone.

| Model | FLOPs | Accuracy Top-1 (%) | Top-5 (%) |
|---|---|---|---|
| ResNet-50 [2] | 4.1 G | 76.32 | 92.95 |
| ResNet-50 + Cutout [10] | 4.1 G | 77.07 | 93.34 |
| ResNet-50 + Dropblock [18] | 4.1 G | 78.13 | 94.02 |
| ResNet-50 + Mixup [12] | 4.1 G | 77.9 | 93.9 |
| ResNet-50 + CutMix [13] | 4.1 G | 78.60 | 94.08 |
| ResNet-50 + StochDepth [15] | 4.1 G | 77.53 | 93.73 |
| ResNet-50 + Droppath [16] | 4.1 G | 77.10 | 93.50 |
| ResNet-50 + ShakeDrop [22] | 4.1 G | 77.5 | - |
| ResNet-50 + GradAug (Ours) | 4.1 G | **78.79** | **94.38** |
| ResNet-50 + bag of tricks [28] | 4.3 G | 79.29 | 94.63 |
| ResNet-50 + GradAug† (Ours) | **4.1 G** | **79.67** | **94.93** |

# 4 Experiments

We first evaluate the effectiveness of GradAug on image classification. Next, we show the generalization ability of GradAug on object detection and instance segmentation. Finally, we demonstrate that GradAug can improve the model's robustness to image distortions and adversarial attacks. We also show GradAug is effective in low data settings and can be extended to semi-supervised learning.

## 4.1 ImageNet classification

**Implementation details.** ImageNet [27] dataset contains 1.2 million training images and 50,000 validation images in 1000 categories. We follow the same data augmentations in [13] to have a fair comparison. On ResNet-50, we train the model for 120 epochs with a batch size of 512. The initial learning rate is 0.2 with cosine decay schedule. We sample $n = 3$ sub-networks in each training iteration and the width lower bound is $\alpha = 0.9$. For simplicity, we only use random scale transformation for sub-networks. That is the input images are randomly resized to one of $\{224 \times 224, 192 \times 192, 160 \times 160, 128 \times 128\}$. Note that we report the final-epoch accuracy rather than the highest accuracy in the whole training process as is reported in CutMix [13].

We evaluate GradAug and several popular regularization methods on the widely used ResNet-50 [2]. The results are shown in Table 1. GradAug achieves a new state-of-the-art performance of 78.79% based on ResNet-50. Specifically, GradAug significantly outperforms the structure regularization methods by more than 1 point. As illustrated in Eq. 3 and Eq. 4, GradAug has a similar form with structure regularization. The difference is that GradAug introduces self-guided disturbances to augment the raw gradients. The large improvement over the structure regularization methods clearly validates the effectiveness of our proposed method.

As shown in Eq. 6, GradAug can be seamlessly combined with data augmentation. We combine GradAug with CutMix (p=0.5) and denote this method as GradAug†. We compare GradAug† with bag of tricks [28] at the bottom of Table 1. It is evident that GradAug† outperforms bag of tricks both in model complexity and accuracy. Note that bag of tricks includes a host of advanced techniques such as model tweaks, training refinements, label smoothing, knowledge distillation, Mixup augmentation, etc., while GradAug is as easy as regular model training.

Due to the sub-networks in GradAug training, one natural question arises: *Would the training cost of GradAug increase significantly?* As stated in [13], typical regularization methods [12, 13, 18] require more training epochs to converge, while GradAug converges with less epochs. Thus the total training time is comparable. The memory cost is also comparable because sub-networks do forward and back-propagation one by one, and only their gradients are accumulated to update the weights. Table 2 shows the comparison on ImageNet. The training cost is measured on an $8 \times$ 1080Ti GPU server with a batch size of 512. We can see that the training time of GradAug is comparable with state-of-the-art regularization methods such as CutMix.

Table 2: Training cost of state-of-the-art regularization methods on ImageNet.

| ResNet-50 | #Epochs | Mem (MB) | Mins/epoch | Total hours | Top-1 Acc (%) |
|---|---|---|---|---|---|
| Baseline [12] | 90 | 6973 | 22 | **33** | 76.5 |
| Baseline [12] | 200 | 6973 | 22 | 73 | 76.4 |
| Mixup [12] | 90 | 6973 | 23 | 35 | 76.7 |
| Mixup [12] | 200 | 6973 | 23 | 77 | 77.9 |
| Dropblock [18] | 270 | 6973 | 23 | 104 | 78.1 |
| CutMix [13] | 300 | 6973 | 23 | 115 | 78.6 |
| GradAug | 120 | 7145 | 61 | 122 | **78.8** |
| GradAug | 200 | 7145 | 61 | 203 | **78.8** |

Table 3: Cifar-100 classification accuracy of different techniques on WideResNet and PyramidNet.

| Model | WideResNet-28-10 | | PyramidNet-200 ($\tilde{\alpha} = 240$) | |
|---|---|---|---|---|
| | Top-1 Acc (%) | Top-5 Acc (%) | Top-1 Acc (%) | Top-5 Acc (%) |
| Baseline [2] | 81.53 | 95.59 | 83.49 | 94.31 |
| + Mixup [12] | 82.5 | - | 84.37 | 96.01 |
| + CutMix [13] | **84.08** | **96.28** | 84.83 | 86.73 |
| + ShakeDrop [22] | 81.65 | 96.19 | 84.57 | 97.08 |
| + GradAug (Ours) | 83.98 | 96.17 | **84.98** | **97.08** |
| + CutMix + ShakeDrop | 81.64 | 96.46 | 85.93 | **97.63** |
| + GradAug† (Ours) | **85.25** | **96.85** | **86.24** | 97.33 |

## 4.2 Cifar classification

**Implementation details.** We also evaluate GradAug on Cifar-100 dataset [29]. The dataset has 50,000 images for training and 10,000 images for testing in 100 categories. We choose WideResNet [30] and PyramidNet [23] structures as they achieve state-of-the-art performance on Cifar dataset. We follow the training setting in [23,30] in our experiments. For WideResNet, we train the model for 200 epochs with a batch size of 128. The initial learning rate is 0.1 with cosine decay schedule. Weight decay is 0.0005. PyramidNet is trained for 300 epochs with a batch size of 64. The initial learning rate is 0.25 and decays by a factor of 0.1 at 150 and 225 epochs. Weight decay is 0.0001. We use random scale transformation where input images are resized to one of $\{32 \times 32, 28 \times 28, 24 \times 24\}$. The number of sub-networks is $n = 3$ and the width lower bound is $\alpha = 0.8$.

The results are compared in Table 3. GradAug is comparable with the state-of-the-art CutMix, and it clearly outperforms the best structure regularization method ShakeDrop, which validate the effectiveness of the self-guided augmentation to the raw gradients. We further illustrate this by comparing GradAug† with CutMix + ShakeDrop. On WideResNet, ShakeDrop severely degrades the Top-1 accuracy of CutMix by 2.44%, while GradAug consistently improves CutMix by more than 1 point. The reason is that ShakeDrop introduces random noises to the training process, which is unstable and ineffective in some cases. However, GradAug is a self-guided augmentation to the gradients, which makes it compatible with various structures and data augmentations.

## 4.3 Ablation study

We study the contribution of random width sampling and random transformation to the performance, respectively. We also show the impact of the number of sub-networks $n$ and the width lower bound $\alpha$. The experiments are conducted on Cifar-100 based on the WideResNet-28-10 backbone.

**Random width sampling and random transformation.** We study the effect of one component by abandoning the other one. First, we do not randomly sample sub-networks. Then GradAug becomes multi-scale training in our experiments. In each iteration, we feed different scaled images to the network. Second, we do not conduct random scale transformation. In each iteration, we sample 3 sub-networks and feed them with the original images. The results are shown in Table 4. Random scale and random width sampling only achieve marginal improvements over the baseline, but GradAug

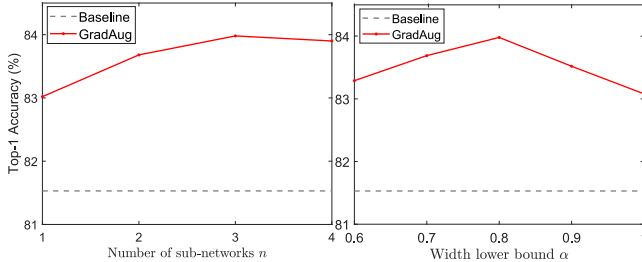

Figure 2: Effect of number of sub-networks and width lower bound.

Table 4: Contribution of random width sampling and random scale on Cifar-100.

| WideResNet -28-10 | Top-1 Acc | Top-5 Acc |
|---|---|---|
| Baseilne | 81.53 | 95.59 |
| RandScale | 82.27 | 96.16 |
| RandWidth | 81.74 | 95.56 |
| GradAug | 83.98 | 96.17 |

Table 5: Top-1 Accuracy (%) of WideResNet-28-10 on Cifar-100 and ResNet-50 on ImageNet.

| Model | WideResNet | ResNet |
|---|---|---|
| Baseline | 81.53 | 76.32 |
| RandScale | 83.98 | 78.79 |
| RandRot | 83.36 | 77.62 |
| Scale&Rot | 84.21 | 78.66 |

Table 6: Utilizing stochastic depth in GradAug. We list the top-1 accuracy reported in the paper and our re-implementation.

| ResNet-110 | Cifar-10 | | Cifar-100 | |
|---|---|---|---|---|
| | Reported | Reimpl. | Reported | Reimpl. |
| Baseline | 93.59 | 93.49 | 72.24 | 72.21 |
| StochDepth [15] | 94.75 | 94.29 | 75.02 | 75.20 |
| GradAug | - | **94.85** | - | **77.01** |

remarkably enhances the baseline (+2.43%). This reaffirms the effectiveness of our method, which unifies data augmentation and structure regularization in the same framework for better performance.

**Number of sub-networks and width lower bound.** There are two hyperparameters in GradAug, the number of sub-networks $n$ and sub-network width lower bound $\alpha$. We first explore the effect of $n$. Other settings are the same as Section 4.2. The results are shown in Figure 2. A larger $n$ tends to achieve higher performance since it involves more self-guided gradient augmentations. The accuracy plateaus when $n \geq 3$. Note that even one sub-network can significantly improve the baseline. Then we investigate the impact of width lower bound $\alpha$ by fixing other settings. As shown in Figure 2, $\alpha = 0.8$ achieves the best accuracy, but all the values clearly outperform the baseline. GradAug is not sensitive to these hyperparameters. Empirically, we can set $n \geq 3$ and $\alpha \in [0.7, 0.9]$.

**Effect of different transformations.** As shown in experiments above, GradAug is very effective when leveraging random scale transformation and CutMix. Here we further explore other transformations, including random rotation transformation and the combination of random scale and rotation transformations. We conduct the experiments on WideResNet-28-10 and ResNet-50 following the settings above. For random rotation, we randomly rotate the images by a degree of $\{0°, 90°, 180°, 270°\}$. For the combination, the input images are first randomly rotated and then randomly resized. The results are shown in Table 5. It is clear that both transformations (random scale and random rotation) and their combination achieve significant improvements over the baseline. This validates our idea of regularizing sub-networks by different transformed images.

**Generating sub-networks by stochastic depth.** In the experiments above, we generate sub-networks by cutting the network width. Similarly, we can generate sub-network by shrinking the network depth. We follow StochDepth [15] to randomly drop some layers during training. The training settings are the same as [15] and we use random scale transformation to regularize sub-networks. As shown in Table 6, GradAug significantly outperforms the baseline and StochDepth. This demonstrates that GradAug can be generalized to depth-shortened sub-networks and again verifies the effectiveness of our idea.

## 4.4 Object detection and instance segmentation

To evaluate the generalization ability of the learned representations by GradAug, we finetune its ImageNet pretrained model for COCO [31] object detection and instance segmentation. The experiments are based on Mask-RCNN-FPN [6, 32] framework and MMDetection toolbox [33] on ResNet-50 backbone. Mixup and CutMix, two most effective methods in image classification, are employed for comparison. As explained in Section 2, Mixup and CutMix are mixed sample data augmentation methods, which can not be applied to object detection and segmentation. Therefore, we compare these methods by directly finetuning their ImageNet pretrained models on COCO dataset.

Table 7: COCO object detection and instance segmentation based on Mask-RCNN-FPN.

| Model | ImageNet Cls Acc (%) | Det mAP | Seg mAP |
|---|---|---|---|
| ResNet-50 (Baseline) | 76.3 (+0.0) | 36.5 (+0.0) | 33.3 (+0.0) |
| Mixup-pretrained | 77.9 (+1.6) | 35.9 (-0.6) | 32.7 (-0.6) |
| CutMix-pretrained | 78.6 (+2.3) | 36.7 (+0.2) | 33.4 (+0.1) |
| GradAug-pretrained | **78.8 (+2.5)** | **37.7 (+1.2)** | **34.5 (+1.2)** |
| GradAug | **78.8 (+2.5)** | **38.2 (+1.7)** | **35.4 (+2.1)** |

All models are trained with $1\times$ schedule on COCO dataset. The image resolution is $1000 \times 600$. The mean Average Precision (AP at IoU=0.50:0.05:0.95) is reported in Table 7. We can see that although Mixup and CutMix achieve large improvements on ImageNet classification, the learned representations can barely benefit object detection and segmentation. In contrast, GradAug-pretrained model considerably improves the performance of Mask-RCNN. This validates that GradAug enables the model to learn well-generalized representations which transfer well to other tasks.

Moreover, the training procedure of GradAug can be directly applied to the detection framework. The result (last line of Table 7) shows that it further boosts the performance as compared with GradAug-pretrained and can significantly improve the baseline by +1.7 det mAP and +2.1 seg mAP. The implementation details and qualitative results are in the **supplementary material**.

## 4.5 Model robustness

Deep neural networks are easily fooled by unrecognizable changes on input images. Developing robust machine learning models is pivotal for safety-critical applications. In this section, we evaluate the model robustness to two kinds of permutations, image corruptions and adversarial attacks.

Table 8: Corruption error of ResNet-50 trained by different methods. **The lower the better**.

| Model | Clean Err | Noise | | | Blur | | | | Weather | | | | Digital | | | | mCE |
|---|---|---|---|---|---|---|---|---|---|---|---|---|---|---|---|---|---|
| | | Gauss | Shot | Impulse | Defocus | Glass | Motion | Zoom | Snow | Frost | Fog | Bright | Contrast | Elastic | Pixel | JPEG | |
| ResNet-50 | 23.7 | 72 | 75 | 76 | 77 | 91 | 82 | 81 | 78 | 76 | 65 | 59 | 65 | 89 | 72 | 75 | 75.5 |
| + Cutout | 22.9 | 72 | 74 | 77 | 77 | 91 | 80 | 80 | 77 | 77 | 65 | 58 | 64 | 89 | 75 | 76 | 75.5 |
| + Mixup | 22.1 | 68 | 72 | 72 | **75** | **88** | **75** | **74** | 70 | 70 | 55 | 55 | 61 | **85** | 65 | 72 | 70.5 |
| + CutMix | 21.4 | 72 | 74 | 76 | 77 | 91 | 78 | 78 | 77 | 75 | 62 | 56 | 65 | 87 | 77 | 74 | 74.6 |
| + GradAug | 21.2 | 72 | 72 | 79 | 78 | 90 | 80 | 80 | 73 | 72 | 61 | 55 | 64 | 87 | **64** | 71 | 73.2 |
| + GradAug† | **20.4** | 71 | 73 | 78 | 76 | 91 | 78 | 77 | 72 | 71 | 61 | **53** | 63 | 86 | 76 | **69** | 73.0 |
| + GradAug* | 21.9 | 62 | 65 | 63 | 77 | 90 | 79 | 75 | **64** | **57** | **50** | 54 | **52** | 87 | 77 | 75 | **68.5** |

**Image corruption.** ImageNet-C dataset [34] is created by introducing a set of 75 common visual corruptions to ImageNet classification. ImageNet-C has 15 types of corruptions drawn from four categories (noise, blur, weather and digital). Each type of corruption has 5 levels of severity. Corruptions are applied to validation set only. Models trained on clean ImageNet should be tested on the corrupted validation set without retraining. We follow the evaluation metrics in [34] to test ResNet-50 trained by different regularization methods. The mean corruption error (mCE) is reported in Table 8. Mixup has lower mCE than other methods. We conjecture the reason is that Mixup proportionally combines two samples, which is in a similar manner to the generation of corrupted images. GradAug outperforms the second best competing method CutMix by 1.4%. Note that GradAug can also be combined with Mixup and we denote it as GradAug*. The results in Table 8 reveal that GradAug* further improves Mixup and achieves the lowest mCE. This demonstrates that GradAug is capable of leveraging the advantages of different augmentations.

**Adversarial attack.** We also evaluate model robustness to adversarial samples. Different from image corruption, adversarial attack uses a small distortion which is carefully crafted to confuse a classifier. We use Fast Gradient Sign Method (FGSM) [35] to generate adversarial distortions and conduct white-box attack to ResNet-50 trained by different methods. The classification accuracy on adversarially attacked ImageNet validation set is reported in Table 9. Note that here Mixup is not as robust as to image corruptions, which validates our aforementioned conjecture in the image corruption experiment. GradAug and CutMix are comparable and both significantly outperform other methods. GradAug† further gains improvements over GradAug and CutMix, manifesting superiority of our self-guided gradient augmentation.

Table 9: ImageNet Top-1 accuracy after FGSM attack. $\epsilon$ is the attack severity.

| Model | $\epsilon = 0.05$ | $\epsilon = 0.10$ | $\epsilon = 0.15$ | $\epsilon = 0.20$ | $\epsilon = 0.25$ |
|---|---|---|---|---|---|
| ResNet-50 | 27.90 | 22.65 | 19.50 | 17.04 | 15.09 |
| + Cutout | 27.22 | 21.55 | 17.51 | 14.68 | 12.37 |
| + Mixup | 30.76 | 25.59 | 21.63 | 18.44 | 16.19 |
| + CutMix | 37.73 | 33.42 | 29.69 | 26.29 | 23.26 |
| + GradAug | 36.51 | 31.44 | 27.70 | 24.93 | 22.33 |
| + GradAug† | **40.26** | **35.18** | **31.36** | **28.04** | **25.12** |

Table 10: Top-1 accuracy on Cifar-10 and STL-10 with limited labels.

| Model | Cifar-10 | | | STL-10 |
|---|---|---|---|---|
| | 250 | 1000 | 4000 | 1000 |
| WideResNet-28-2 | 45.23±1.01 | 64.72±1.18 | 80.17±0.68 | 67.62±1.06 |
| + CutMix (p=0.5) | 43.45±1.98 | 63.21±0.73 | 80.28±0.26 | 67.91±1.15 |
| + CutMix (p=0.1) | 43.98±1.15 | 64.60±0.86 | 82.14±0.65 | 69.34±0.70 |
| + ShakeDrop | 42.01±1.94 | 63.11±1.22 | 79.62±0.77 | 66.51±0.99 |
| + GradAug | **50.11±1.21** | **70.39±0.82** | **83.69±0.51** | **70.42±0.81** |
| + GradAug-semi | **52.95±2.15** | **71.74±0.77** | 84.11±0.25 | **70.86±0.71** |
| Mean Teacher [36] | 48.41±1.01 | 65.57±0.83 | **84.13±0.28** | - |

## 4.6 Low data setting

Deep neural network models suffer from more severe over-fitting when there is only limited amount of training data. Thus we expect regularization methods to show its superiority in low data setting. However, we find that state-of-the-art methods are not as effective as supposed. For a fair comparison, we follow the same hyperparameter settings in [37]. The backbone network is WideResNet-28-2. We first evaluate different methods on Cifar-10 with 250, 1000 and 4000 labels. Training images are sampled uniformly from 10 categories. We run each model on 5 random data splits and report the mean and standard deviation in Table 10. We observe that CutMix (p=0.5) and ShakeDrop even degrade the baseline model performance, especially when labels are very limited. CutMix mixes images and their labels, which introduces strong noises to the data and ground truth labels. This is effective when there is enough clean labels to learn a good baseline. But when the baseline is weak, this disturbance is too severe. We reduce the impact of CutMix by setting p=0.1, where CutMix is barely used during training. CutMix still harms the baseline when there are only 250 labels, but it becomes beneficial when there are 4000 labels. ShakeDrop has a similar trend with CutMix since it introduces noises to the structure. In contrast, GradAug significantly and consistently enhances the baseline in all cases because it generates self-guided augmentations to the baseline rather than noises. Moreover, GradAug can be easily extended to semi-supervised learning (SSL). We can leverage the full-network to generate labels for unlabeled data and use them to train the sub-networks. See the **supplementary material** for implementation details. Our GradAug-semi can further improve the performance over GradAug. It even achieves comparable performance with Mean Teacher [36], which is a popular SSL algorithm. We also evaluate the methods on STL-10 dataset [38]. The dataset is designed to test SSL algorithms, where the unlabeled data are sampled from a different distribution than labeled data. Similarly, CutMix and ShakeDrop are not effective while GradAug and GradAug-semi achieve clear improvements.

## 5 Conclusion

In this paper, we propose GradAug which introduces self-guided augmentations to the network gradients during training. The method is easy to implement while being effective. It achieves a new state-of-the-art accuracy on ImageNet classification. The generalization ability is verified on COCO object detection and instance segmentation. GradAug is also robust to image corruption and adversarial attack. We further reveal that current state-of-the-art methods do not perform well in low data setting, while GradAug consistently enhances the baseline in all cases.

## Acknowledgments

This work is partially supported by the National Science Foundation (NSF) under Grant No. 1910844 and NSF/Intel Partnership on MLWiNS under Grant No. 2003198.

## Broader Impact

The proposed regularization method is a generic approach for deep neural networks training. Researchers in the machine learning and computer vision communities should benefit from this work. To the best of our knowledge, we don't think this research will put anyone at disadvantage. All the experiments are based on the public datasets and follow the standard experimental settings. Thus the method does not leverage biases in the data.

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
