[Supplementary Material]

# Supplementary Material of
# GradAug: A New Regularization Method for Deep Neural Networks

**Taojiannan Yang, Sijie Zhu, Chen Chen**
University of North Carolina at Charlotte
{tyang30,szhu3,chen.chen}@uncc.edu

## Abstract

The supplementary material includes the following items.

1. The effect of the two training tricks.
2. Implementation details on object detection and low data setting experiments.
3. Visual examples of object detection and instance segmentation.

## 1   Effect of the training tricks

In GradAug, we adopted two training tricks, smallest sub-network (SS) and soft label (SL), to further improve the performance. Here we study the effect of these two tricks. Ablation is in Table 1. SS does not make a big difference, and due to the limit of time, we didn't conduct the experiment of SS on ImageNet. SL is important in GradAug, but the application of SL is not trivial. First, soft labels come for free (from the full network) in GradAug, whether sampling sub-networks by width or depth. Second, our GradAug framework transfers the knowledge among sub-networks based on **differently transformed inputs**. This is different from traditional knowledge distillation (KD) where a strong teacher is required and the transferred knowledge is based on the same input. As shown in [1], normal KD usually marginally improves the performance on ImageNet, but the large improvement in GradAug actually validates our idea of regularizing sub-networks with different inputs.

Table 1: The effect of smallest sub-network (SS) and soft label (SL). Top-1 accuracy is reported.

| Model | Cifar-100 | ImageNet |
|---|---|---|
| Baseline | 81.5 | 76.3 |
| GradAug | 84.0 | 78.8 |
| no SS | 83.8 | - |
| no SS&SL | 82.5 | 77.4 |

## 2   Implementation details

### 2.1   Object detection and instance segmentation

The experiments are based on Mask-RCNN-FPN [2, 3] using the ResNet-50 [4] backbone. The implementation is based on MMDetection [5]. For the experiments which only use the pretrained models, we replace the backbone network with ResNet-50 trained by different methods on ImageNet. Then we train the detection framework on COCO following the default settings in MMDetection. The only difference is that we use $1000 \times 600$ image resolution rather than $1333 \times 800$ to reduce memory cost.

We also show that GradAug can be applied to the detection framework. Following the setting in ImageNet classification, the number of sub-networks $n$ is 4 and the width lower bound $\alpha$ is 0.9. We also use the simple random scale transformation. The shorter edge of the input image is randomly resized to one of $\{600, 500, 400, 300\}$ while keeping the aspect ratio. We only do sub-network sampling on the network backbone. The FPN neck and detection head are shared among different sub-networks. For simplicity, sub-networks are trained with the ground truth rather than soft labels. Other settings are the same as the default settings of $1\times$ training schedule in MMDetection.

## 2.2 Low data setting

In this experiment, we follow the settings in [6] to facilitate a fair comparison. We evaluate different methods using WideResNet-28-2 on Cifar-10 with 250, 1000 and 4000 labels. All models are trained only on the labeled data by Adam optimizer [7] for 500,000 iterations, with a batch size of 50. The initial learning rate is 0.001 and decays by 0.2 at 200,000 and 400,000 iterations. GradAug uses the same hyperparameters as in Cifar-100 experiment. The number of sub-networks $n$ is 4 and the width lower bound $\alpha$ is 0.8. Input images are randomly resized to one of $\{32 \times 32, 28 \times 28, 24 \times 24\}$.

We also demonstrate that GradAug can easily leverage unlabeled data. For each unlabeled image, we first feed it to the full-network and use the output of the full-network as its pseudo-label. Then we can leverage unlabeled images and their pseudo-labels to train sub-networks as stated in GradAug. The batch size for unlabeled images is 50. Other settings are the same as in fully-supervised experiments.

## 3 Visual examples of object detection and instance segmentation

In this section, we show some visual results of object detection and instance segmentation by using the baseline model MaskRCNN-R50 and the model trained by GradAug. As depicted in Figure 1, GradAug-trained model can effectively detect small scale and large scale objects. It is also robust to occlusions. This indicates that GradAug helps the model learn well-generalized representations.

Figure 1: Object detection and instance segmentation visual examples.