[Reviews · NeurIPS 2020]

Review 1

Summary and Contributions: The paper proposes a new regularization call GradAug to better train deep neural networks. The main contributions are: 1. A multi forward method of different data augmentations using different sub-networks is proposed. And the method is viewed as gradient augmentation technique by the authors. 2. Extensive experiments on image classification, object detection, instance segmentation, adversarial attack and low data setting are conducted to demonstrate the effectiveness of the proposed method.

Strengths: 1. The authors proposes to regularize neural nets by forward and backward the combination of multiple data augmentations. And different data augmentations go into different sub-networks, and the final sub-network sampling is simply keeping the first w (w\in[0,1]) percent of the total filters and outputs channels of each layer. The whole procedure is introduced as a special gradient augmentation. 2. The authors show the improvements of the proposed methods in many different tasks and datasets. Beyond the accuracy of recognition, detection and segmentation, the model robustness to adversarial samples are also improved.

Weaknesses: 1. The training time and memory cost could increase by several times. 2. Since the training is more time consuming, many other solutions such as mimicking / distilling based solutions are suggested to compare with. 3. Only a very simple sub-network sampling strategy is considered, what about randomly choosing a sub-network each time, or keeping the most part fixed and a some portion randomly chosen? 4. Only a very simple data augmentation (different input training size) is considered in the sub-network training. What about other choices? ====== post rebuttal ======= I misunderstood some important points in the paper in my original review comments. 1. The difference between GradAug and GradAug+. Now that I know that GradAug+ (GradAug with CutMix augmentations) does achieve STOA results on several tasks. For example, for ImageNet classification, the result of 79.6% acc of top 1 is STOA as I far as I know. 2. Experiments show that GradAug need less epochs to converge to a good result, and this alleviate my concern about the time cost to some extent. 3. The idea also seems to work good in the setting of stochastic depth.

Correctness: The method looks reasonable, but more experiments such as different sub-network sampling and data augmentation strategies are suggested.

Clarity: the paper is good written and easy to understand.

Relation to Prior Work: The relate work is clearly introduced and compared with.

Reproducibility: Yes

Additional Feedback:


Review 2

Summary and Contributions: After rebuttal and discussion with other reviewers I have updated my score. However, I do point out several concerns of mine which the authors could consider further validation for: It's good that the authors performed the time/memory comparison in the rebuttal as that was a significant concern of mine. My concerns mostly revolve around what other techniques should we compare this against? Given that this algorithm takes 3-4x the time with comparison to the baseline, I could for example: 1: Train a much larger network and then use compression techniques to slim it to the same size. 2: Train an even larger network at a lower quantization then use compression. 3: Train a larger network with competing regularization techniques (e.g. Mixup which is still ~70% faster. see rebuttal) then use compression. Is it sufficiently true that this approach is in fact *not* aliasing the idea of training a larger network, then compressing it? Is it sufficiently true that the novelty of the approach (i.e. subnetwork learning) cannot be captured by training a larger network, then compressing? These questions would be nice to be answered with convincing validation, and my score would certainly be higher if they were validated. This paper proposes a novel form of regularization dubbed 'GradAug'. GradAug consists of two components. One is structured subsampling of neural networks similar to dropout. A second is a 'self guided gradient augmentation' technique is also used. GradAug iterates upon previous work such as dropout/zoneout/freezeout to improve the robustness of neural networks by incorporating training them as 'ensembles' of smaller subnetworks. Some insight is provided into the underlying operating mechanism behind GradAug. Validation is also performed on GradAug demonstrating improved test performance on CIFAR10/Imagenet. The general applicability of the technique is demonstrated by testing on other image tasks. Finally, it is demonstrated that GradAug improves the robustness of the network to adversarial attacks and image corruption.

Strengths: The empirical results shown by the approach are quite strong, and certainly demonstrate the possible validity of the approach. There is some amount of novelty in the approach, however it builds upon previous work considering a network as a an ensemble of smaller subnetworks (i.e. dropout et. al.).

Weaknesses: There remain several weaknesses in the paper, mostly centering around thoroughness of validation, and inadequate comparison to other (semi-related) work. In particular I found the explanation as to how the two proposed approaches in GradAug. It is not sufficiently explained (e.g. in 3.2) *how* the structured subnetwork sampling, as well as the 'self guided gradient augmentation' aid in building a robust network. Although I agree the empirical results are impressive, I'm not sure how this approach works outside of imagining it as some sort of 'Dropout, but better,' technique. From what I can tell the 'self guided gradient augmentation' prefers networks which are robust (i.e. preferring little to no change in the output, given scaling/transformations according to Eq. 4) to certain types of transformations (e.g. scaling, rotation, translation etc.). This should improve robustness, however the authors claim this is some sort of 'self guiding'. It's not exactly clear what is meant by this. The authors need to better explain exactly what is hoped to be achieved by the self guiding, and also to furnish evidence through empirical experiments, or perhaps proof/symbolic intuition. It is also not fully well explored how the structured subnetwork training improves upon Dropout, and how it aids in training and regularizing networks. The authors mention a 'a larger sub-network always share the representations of a smaller sub-network in a weights-sharing training fashion, so it can leverage the representations learned in smaller sub-networks.' What is precisely meant by this? I'm not sure I understand, or believe that such an assertion is true. This is not elaborated upon in later sections or empirically in the validation. Two additional weaknesses in the paper are a lack of mention/evaluation on the performance cost of GradAug. From what I can tell, GradAug requires several forward passes for each step of training. How significantly does this affect memory/training time? This would have been helpful to help evaluate this work, as typical regularization methods are usually cheap to apply. In the proposed approach, several forward passes are made using subnetworks to provide gradient for a single backward pass. This can be thought of as distilling the knowledge of a more expressive, powerful network into a smaller one. In this way this approach can be viewed somewhat analogous to neural network compression. Although this comparison is not perfect, it possibly deserves further comparison. Overall, this paper, though shows strong empirical results, has not fully or well explored its proposed approach. Due to this reason, I don't believe it is ready for publication.

Correctness: Overall I agree with the central claim that GradAug improves performance and robustness as demonstrated. However the mechanism of action remains unclear, and is not well validated in the paper. Thus some of the claims regarding how GradAug improves robustness/generalization are not well validated.

Clarity: The paper is written 'ok'. Certainly some of it can be improved using clearer text. In particular I found the assertions and claims made in 3.2 difficult to understand. What does it mean for the 'disturbances' to be self guiding? I'm not sure what I should take away from this section as to the overall operating mechanism of GradAug as it is not clearly stated.

Relation to Prior Work: This is mostly done, however I have some concerns which I have elaborated upon in the weaknesses (see second to last paragraph).

Reproducibility: Yes

Additional Feedback:


Review 3

Summary and Contributions: This paper proposed a new regularization method that leverages differently transformed input to regularize a set of sub-networks originated from the full-network. The idea is that sub-networks should recognize transformed images as the same objects with the full-network. The author analyzes its effect from the gradient view and conducted thorough experiments to validate the idea. The method is demonstrated to outperform state-of-the-art methods on different tasks.

Strengths: 1.The idea of leveraging different transformed images to regularize the sub-networks and thereby providing self-guided augmentation to the gradients is novel and interesting. It is simple to implement and can be combined with other regularization schemes. Overall, the paper is well-presented and the proposed idea is clearly conveyed. 2.The analysis on the gradient property of different methods is reasonable and useful to understand the differences between this work and other regularization techniques. The claim is also validated in the experiments. 3.The experimental results are very promising based on a set of experiments for a range of tasks. Specifically, the authors demonstrate that state-of-the-art methods can hardly improve in downstream tasks and are not effective in low data setting, while the proposed method shows effectiveness in these tasks.

Weaknesses: 1.The author may need more experiments to show the effect of different transformations. Though in the supplementary the authors experimented with random rotation and the combination of random scale and rotation, more experiments on different models and larger datasets will make it more convincing. There might be space issue, but I think this experiment should also be put in the main paper rather than the supplementary. 2. The claim on model robustness to adversarial attack may be too strong. FGSM is just one type of adversarial attack approach. To claim the general model robustness, the authors may need more experiments on different adversarial attacks. While I can understand that the proposed method is not focused on adversarial attack, it still would be more precise to claim the robustness to FGSM attack based on the experiments in the paper. 3. Although it is stated that the training procedure of the proposed GradAug is similar to the regular network, the training time for each epoch might be longer due to sub-networks. It would be good to include such discussion and analysis. 4. In ImageNet classification experiment, the images are randomly resized to one of {224, 192, 160, 128}. It is not clear why these particular resolutions are selected. How the image resolution and the number of resolutions that the sub-networks can choose would affect the performance?

Correctness: The method is clear and straightforward and appears correct. However, as mentioned in the weakness section, the claim on model robustness to adversarial attack may be too strong without more experiments on other adversarial attack approaches. It would be more precise to claim the robustness to FGSM attack based on the experiments.

Clarity: The paper is clearly written and well structured, and the figures and algorithm pseudo-code help in the understanding.

Relation to Prior Work: The distinctions between this work and prior research are clearly discussed. In particular, the gradient flow analysis of different regularization techniques presented in Sec. 3.2 provides strong evidence on the differences.

Reproducibility: Yes

Additional Feedback: =====post-rebuttal=======: After reading through the rebuttal and considering the other reviewer's comments, I again feel this paper is a very good submission. I found the work to be clear and well-reasoned, and the results to be impressive on various tasks. The most significant contribution would be the idea of regularizing the sub-networks with transformed input, which is new and interesting. In the rebuttal, the authors have run additional experiments demonstrating (1) the training cost (memory and time) of the proposed GradAug is comparable with the state-of-the-art regularization approaches, which is attributed to the faster convergence speed of GradAug; (2) GradAug is able to generalize to sub-networks generated by shrinking the full-network’s depth (R4’s suggestion), which also reveals the flexibility of the framework (another plus of GradAug). I agree the strategy of generating the sub-networks is a good direction to explore further, and it probably deserves a thorough theoretical analysis and experimental investigation, which, in my opinion, is beyond the scope of this paper - describing the general idea and framework. Overall, I think this is a decent research work in terms of the method and experimental evaluation. And developing regularization techniques from both data and network aspects might be worth further exploring.


Review 4

Summary and Contributions: This paper proposes a new regularization method, GradAug. During training, GradAug trains sub-networks with various input transformations, and aggregates the losses generated by the full network and sub-networks.

Strengths: 1. This method significantly boosts image classification and the object detection task.

Weaknesses: 1. Similarity with slimmable network [*] Defining sub-networks using width ratio is quite similar to that of slimmable network, and also many training tricks were adopted from slimmable network paper. Surely, applying data augmentation for each sub-network is different, but this training scheme is not a new thing. How critical the adopted two training tricks (soft labels and smallest sub-network) for the performance of GradAug? How much degrades the accuracy on ImageNet without these tricks? How about applying GradAug upon Slimmable network? Would it works better than or not? Inversely, can GradAug-trained network be pruned as in Slimmable network? [*] Universally Slimmable Networks and Improved Training Techniques, ICCV 2019.

Correctness: Yes

Clarity: Yes

Relation to Prior Work: Yes

Reproducibility: Yes

Additional Feedback: The sub-networks of GradAug are sampled in a similar manner with slimmable network. How about utilizing stochastic depth [*], which randomly drops residual blocks to make depth-shrinked sub networks, to sample sub-nets for GradAug? It would be great if GradAug scheme is generalized like this. [*] Deep Networks with Stochastic Depth, ECCV 2016. ======= Post-rebuttal comment ======== After reading the rebuttal and other reviewers' comments, I would like to incline to keep my original rating (6). I'm happy to see the authors conducting depth-shrinked sub-network experiments, which is one of my major concerns (generalization of the method). I think all the additional experiments in the rebuttal should be included by conducting on ImageNet dataset in the final copy to make the paper more convincing and rigorous.

[Author Response · NeurIPS 2020]

We sincerely thank reviewers for their insightful feedback! We are encouraged that reviewers find our method novel (**R2**,**R3**) and analysis insightful (**R3**). All reviewers (**R1**,**R2**,**R3**,**R4**) agreed that **our method achieved significant improvements in a variety of tasks/settings** (image classification, object detection, instance segmentation, adversarial attack and low data setting) backed with extensive experiments and ablations. We address reviewer comments below.

@**R1**,**R2**,**R3**, Q1: The training cost of GradAug may be several times of typical regularization methods: This is NOT true. As stated in [11], typical regularization methods [11,10,8] require **more training epochs** to converge, while GradAug **converges with less epochs**. Thus the **total training time is comparable**. **The memory cost is also comparable** because we forward and backward sub-networks one by one, only their gradients are ac-cumulated to update the weights. Table 1 shows a comparison on ImageNet. The training cost is measured on an $8\times$ 1080Ti GPU server with a batch size of 512. Mixup and CutMix need 77 and 115 hours to converge, while GradAug converges in 122 hours (120 epochs). So the training cost of our GradAug is comparable with SOTA methods.

Table 1: Training cost comparison on ImageNet. Reference # from paper.

| ResNet-50 | Epochs | Mem (MB) | Mins/epoch | Total hours | Top-1 Acc |
|---|---|---|---|---|---|
| Baseline [10] | 90 | 6973 | 22 | **33** | 76.5 |
| Baseline [10] | 200 | 6973 | 22 | 73 | 76.4 |
| Mixup [10] | 90 | 6973 | 23 | 35 | 76.7 |
| Mixup [10] | 200 | 6973 | 23 | 77 | 77.9 |
| CutMix [11] | 300 | 6973 | 23 | 115 | 78.6 |
| GradAug | 120 | 7145 | 61 | 122 | **78.8** |
| GradAug | 200 | 7145 | 61 | 203 | **78.8** |

@**R4**, Q2: Use stochastic depth [A] to sample depth-shortened sub-nets for GradAug: Great suggestion! To do so, we follow the settings in [A] to randomly drop layers to generate sub-networks. We also utilize ran-dom scale transformation and input images are randomly resized to one of $\{32 \times 32, 28 \times 28, 24 \times 24\}$. The results in Table 2 show that **GradAug can be generalized to depth-shortened sub-networks as well**. This also validates the effectiveness of our idea - regularizing sub-networks with differently transformed inputs.

@**R1**,**R4**, Q3: Only a simple sub-network sampling strategy is considered: Our goal is to show the effectiveness of regularizing sub-networks with different transformed inputs. To form sub-networks, we just follow the most common practice in previous literature to scale down the network by network width. As shown in the response to **Q2**, sampling sub-networks by **depth** is also feasible, and the corresponding results (Table 2) also validate its efficacy.

Table 2: Utilizing stochastic depth [A] in GradAug.
[A] "Deep networks with stochastic depth" ECCV 2016

| ResNet-110 | Cifar-10 | | Cifar-100 | |
|---|---|---|---|---|
| | Reported | Reimpl. | Reported | Reimpl. |
| Baseline [A] | 93.59 | 93.49 | 72.24 | 72.21 |
| StochDepth [A] | 94.75 | 94.29 | 75.02 | 75.20 |
| GradAug | - | **94.85** | - | **77.01** |

Analyzing the effect of different sampling strategies is interesting and we will certainly explore it in future work.

@**R1**, Q4: Only random scale transformation is considered: This is NOT true. In the paper we conducted *random scale* transformation and *random scale + CutMix* (**L185-187**, GradAug+). In the **supplementary material**, we also showed the results of *random rotation* and *random scale + random rotation* (confirmed by **R3**). Here, we further present results on ImageNet (Table 3). As suggested by **R3**, we will put these results in the main paper.

Table 3: Different transforma-tions in GradAug on ImageNet.

| ResNet-50 | Top-1 | Top-5 |
|---|---|---|
| Baseline | 76.32 | 92.95 |
| RandScale | 78.79 | 94.43 |
| RandRot | 77.62 | 93.66 |
| RandScale&Rot | 78.66 | 94.40 |

@**R2**, Q5: How GradAug works: We believe there is a misunderstanding about our method. Our idea is leveraging different transformed inputs to regularize sub-networks which are originated from the full-network. We explain our method from two views. First, intuitively, full-network shares the representations learned by sub-networks because they share weights. We illustrate this by showing the CAMs of sub-network and full-network. **Fig. 1** (in paper) shows that full-network shares the attention map of sub-network and it can also use the other network part, which sub-networks don't have, to learn additional features. So full-network can capture more semantic information than sub-networks (**L106-112**). Second, we explain the differences between GradAug and other regularization methods from the perspective of gradient flow. Dropout and its variants randomly drop some connections. This can be viewed as adding **random noises** to the original gradients as explained in Eqs.(1,2,3). GradAug can also be viewed as adding a term to the original gradients (Eq. 4), but this term is the gradients of sub-networks with different transformed inputs. Since **sub-networks are part of the full-network**, we call this term **"self-guided"**. It reinforces good descent directions, leading to improved performance and faster convergence. Indeed, the experimental results show that it significantly improves the performance over Dropout variants (78.8 vs. 77.5 (Shakedrop) [20], 78.1 (Dropblock) [16]) and converges faster in terms of training epochs (120 vs. 180 [20], 270 [16]).

@**R2**, Q6: Comparison to neural network compression: We do NOT agree our approach is analogous to neural network compression: We do NOT agree our approach is analogous to neural network compression. Our goal is to improve the performance of the full network rather than compressing the network.

@**R4**, Q7: Can GradAug be applied to SlimNet, can GradAug-trained network be pruned like SlimNet? GradAug can be applied to SlimNet by feeding different transformed inputs to different widths. We believe the performance can be improved since the full-network is considerably improved. If we do sub-nets sampling by width, GradAug-trained network can be pruned like SlimNet. For example, the performance of sub-net $width = 0.9\times$ is 77.6% on ImageNet.

@**R4**, Q8: Effect of smallest sub-net (SS) and soft label (SL): Ablation is in Table 4. SL is important in GradAug, but the application of SL is not trivial. First, soft labels come for free (from full-net) in GradAug, whether sampling sub-nets by width or depth. Second, we are transferring the knowledge among sub-nets based on **differently transformed inputs**. This is different from traditional KD and label smoothing which usually marginally improve the performance on ImageNet. The effectiveness actually validates our idea of regularizing sub-nets with different inputs. We'll include these results.

Table 4: Effect of SS and SL.

| Model | C-100 | IN-1K |
|---|---|---|
| Baseline | 81.5 | 76.3 |
| GradAug | 84.0 | 78.8 |
| no SS | 83.8 | - |
| no SS&SL | 82.5 | 77.4 |

@**R3**, Q9: Claim on adversarial robustness. Choice of input scales: We will revise the claim to the robustness to FGSM attack. The input scales are determined empirically. We don't want the images to be too small.

[Meta-Review · NeurIPS 2020]

All four reviewers support acceptance. They found that the claims are well supported by the strong experiments. Therefore, I recommend accept.